# Conundrums in the Management of Febrile Infants under Three Months of Age and Future Research

**DOI:** 10.3390/antibiotics13010088

**Published:** 2024-01-16

**Authors:** Helena Wilcox, Etimbuk Umana, Emmanuelle Fauteux-Lamarre, Roberto Velasco, Thomas Waterfield

**Affiliations:** 1St. Georges University Hospital NHS Foundation Trust, London SW17 0QT, UK; helena.wilcox1@nhs.net; 2Wellcome Wolfson Institute of Experimental Medicine, Queen’s University Belfast, Belfast BT9 7BL, UK; eumana01@qub.ac.uk; 3Emergency Department, Cork University Hospital, Wilton, T12 DC4A Cork, Ireland; emma.fauteux@hse.ie; 4Pediatric Emergency Unit, Hospital Universitari Parc Tauli, Institut d’Investigació i Innovació I3PT, 08208 Sabadell, Spain; robertovelascozuniga@gmail.com

**Keywords:** febrile infant, bacterial infection, urinary tract infection

## Abstract

Febrile infants under three months of age pose a diagnostic challenge to clinicians. Unlike in older children, the rates of invasive bacterial infections (IBIs), such as bacteraemia or meningitis, are high. This greater risk of IBI combined with the practical challenges of assessing young infants results in a cautious approach with many febrile infants receiving parenteral antibiotics “just in case”. However, there is a range of validated tailored care guidelines that support targeted investigation and management of febrile infants, with a cohort identified as lower risk suitable for fewer invasive procedures and observation without parenteral antibiotics. This manuscript outlines five common conundrums related to the safe application of tailored-care guidelines for the assessment and management of febrile infants under three months of age. It also explores future research which aims to further refine the management of febrile infants.

## 1. Introduction

Infants under three months of age are at a greater risk of invasive bacterial infections (IBIs), such as meningitis and bacterial sepsis, than older children, with approximately 1–3% of febrile infants having an underlying IBI [1,2,3,4,5]. In addition, the risk of urinary tract infection (UTI), considered a serious bacterial infection (SBI), is between 9 and 17% in this cohort [1,2,3,4,5]. Unlike older children, very young infants have immature immune systems and have not completed their primary vaccinations. This, in combination with the fact that infants under three months may have few clinical features of severe infection in the early stages, makes their management challenging [2,6]. For these reasons, many clinicians and previous guidelines have adopted a cautious “just in case” approach. There have been advances in diagnostic testing with the introduction of novel inflammatory markers, many of which have been included in current clinical practice guidelines (CPGs) [2,4,7,8,9,10]. However, there are inconsistencies in the availability of inflammatory markers worldwide, and as such, different CPGs are used in different countries [11,12]. This manuscript discusses the differences in practice worldwide and compares different CPGs. We then aimed to identify and present the different approaches between UK, European, and North American practices when applied to five common conundrums for managing febrile infants. In addition, we discuss future research studies that aim to refine management and delineate controversies.

## 2. Methods

This review comprises relevant publications focusing on the aims described above. An extensive search of the literature was undertaken using key terms such as “febrile infant”, “bacterial infection”, “viral infection”, “urinalysis”, “lumbar puncture”, “vaccine”, “clinical practice guidelines”, and “fever”. A wide variety of studies were gathered from the following sources: PubMed, Scopus, Web of Science, Google Scholar, Embase, and Gray literature. References were also searched for relevant publications. The criteria for the selection of articles were based on clinical variation in practice, conundrums faced by clinicians, and areas of future research on the management of febrile infants, as discussed in this review.

## 3. Clinical Practice Guidelines and the Variation in Practice Worldwide

In the UK and Ireland, the vast majority of febrile infants under three months of age undergo routine blood and urine testing, with three quarters of young infants being admitted and treated with parenteral antibiotics [8,13]. It should be noted that national guidelines in the UK recommend blood and urine testing for all febrile infants under three months of age and advise a low threshold for parenteral antibiotic treatment [8]. Internationally, approaches vary with European and North American guidelines seeking to identify a lower risk cohort, through sequential assessment, that are suitable for management in the community without parenteral antibiotics and without the need for lumbar puncture [2,4,10]. 

The use of sequential assessment to identify lower risk young infants relies on access to trained paediatric doctors, access to urinalysis, availability of blood biomarker testing, and low prevalence of invasive bacterial infection [2]. In the last 30 years, the epidemiology of bacterial infections in young infants has changed in high-income countries owing to prenatal group B streptococcal (GBS) screening, immunisation against Streptococcus pneumoniae, and improvements in food safety [2]. As with many diseases, the majority (98%) of young infant deaths from infection worldwide occur in low-resource settings [14]. Little is known about the validity of tailored approaches developed in high-income settings outside this context. The application of the approaches discussed in this article outside of the healthcare systems for which they were designed is not appropriate because of the differing rates of IBIs, differing rates of comorbid diseases, and a lack of access to certain investigations [15,16].

One of the major differences between CPGs in North America, Europe, the UK, and Ireland is the use and availability of certain biomarkers as point-of-care testing, specifically procalcitonin (PCT). PCT has emerged as the most specific biomarker with optimum test characteristics for our clinical situation, given that it rises rapidly in the context of IBI with high sensitivity [17]. However, different CPGs utilise different inflammatory markers, depending on their availability. The Rochester criteria, which have been in place since 1994, use only the total white cell count as a biomarker for risk stratification [18]. This was used as a foundation, and with advances in diagnostics, more recent CPGs have utilised CRP and PCT as part of their sequential assessment [4,10]. A European multicentre retrospective study assessed the use of a “step-by-step approach” which included the use of inflammatory markers (e.g., CRP, procalcitonin), urine dipsticks, and urine and blood culture in evaluating 1123 febrile infants <3 months old [19]. The results showed that a sequential approach including clinical and laboratory markers better identified low-risk patients compared to previously used scores (Rochester criteria and Lab-score) [19]. This was then validated in a multicentre prospective study including 2185 infants less than three months old which compared the step-by-step CPG to the Rochester criteria [10]. A total of 2185 infants were enrolled in the study, and 504 (23.1%) were diagnosed with bacterial infections (3.9% had IBIs) [10]. For the identification of IBIs, the step-by-step approach had a sensitivity of 92% and a negative predictive value (NPV) of 99.3%, compared with 81.6% and 98.3%, respectively, using the Rochester criteria [10]. In the low-risk group, seven patients with IBI were missed using the step-by-step approach [10]. It is worth noting that six of the seven IBIs missed had fevers for <2 h, suggesting caution in infants presenting early in their febrile illness [10].

The Paediatric Emergency Care Combined Research Network (PECARN) rule uses urinalysis, absolute neutrophil count, and PCT for risk stratification of well-appearing infants between 0 and 60 days of age [20]. The PECARN rule had a sensitivity of 97.7 (in a derivation study) and 98.8% (in a validation study) [20]. The NPV for serious bacterial infection (SBI) was 99% [20]. With the PECARN rule, three infants were misclassified as low risk, with all three being over 28 days of age. Among them, one had bacteraemia and two had UTIs. 

The American Academy of Pediatrics (AAP) guidelines utilise PCT and ANC or CRP; however, if PCT is unavailable, then temperature >38.5 °C along with ANC and CRP can be utilised [21]. The AAP CPG has been validated in a Canadian cohort without PCT [21]. This prospective diagnostic study included a total of 957 infants [21]. It found that 27 had IBIs (2.8%) and 164 had SBIs (17.1%) [21]. Among those with IBIs, 22 had bacteraemia and 5 had bacteraemia with meningitis [21]. Using the AAP guidelines without PCT, no cases of IBI were misclassified as low risk, giving a sensitivity of 100.0% (95% CI 87.2–100.0) and NPV of 100.0% (95% CI 99.2–100.0) [21]. The sensitivity dropped to 83.5% for SBIs, as 27 infants with positive urine culture results were classified as low risk [21]. The AAP guidelines separate inflammatory markers from urinalysis, and 18 of the 27 cases would have been picked up because of a positive urinalysis [21].

In the UK and Ireland, the National Institute for Health and Care Excellence (NICE) has two guidelines that can be used: NG51, which advises treating all febrile infants for suspected sepsis, and NG143, which is more nuanced and uses white cell count levels in those aged 1–3 months for risk stratification into high- and low-risk infants [8,22]. The British Society for Antimicrobial Chemotherapy (BSAC) is also available, and it deems infants to be low risk if they are clinically well appearing, with a CRP level of <20 mg/L and negative urinalysis [9].

These guidelines have been validated by Paediatric Emergency Research in the United Kingdom and Ireland (PERUKI). Unsurprisingly, NG51 has a sensitivity of 100% given that all children are treated, but this is to the detriment of its specificity: 0% [13]. It is worth acknowledging that although no infant with sepsis is missed with this approach, there are likely to be negative long-term effects, as altering the infant microbiota is linked to the development of atopy and other chronic conditions [13]. This approach also poses questions regarding the overuse of broad-spectrum antibiotics, as links have been made with drug resistance in the newborn period [23]. One study showed that similar antimicrobial resistance patterns to *E. Coli* were found in infants <2 months compared to older children, highlighting the need for antimicrobial stewardship [24].

NG143 had a sensitivity of 91% and specificity of 9%, whereas BSAC had a sensitivity of 82% and specificity of 14% [13]. Figure 1 summarises the different CPGs and their diagnostic accuracy.

## 4. The Variation in Invasive Bacterial Infection Rate According to Age

The risk of IBI decreases with age. Data from a PECARN study of 4778 infants demonstrated that the highest rates of IBI were seen in infants under two weeks of age (5.3%), and that rates of IBI fall steadily by three weeks of age (3.3%) and by four weeks of age (1.6%) in North America [2]. This was also noted in a multicentre North American study including 3066 infants less than 3 months old that had temperatures >38.5 °C measured during their visit [25]. Well-appearing infants aged ≥ 25 days with a temperature ≤ 38.6 °C had a low IBI rate (bacteraemia/meningitis) [25]. For these reasons, all guidelines from the UK, Europe, and North America treat any infant under three weeks of age as high risk for IBI. This is one of the few areas for international consensus. In the UK, Ireland, and Australia, the assessment and management of febrile infants goes up to the 12th week of life. This is in part due to the risk of IBI being similar in those aged 29–60 days (2.2%) and 61–90 days (2.0%) according to UK and Irish data, with urine and blood inflammatory markers recommended as a minimum in this cohort [26,27]. It should be noted that preterm infants and infants with comorbidities were excluded from the studies, and management of these infants should be individualised.

## 5. Febrile Infant Conundrums

Let us now work through five common clinical conundrums, aiming for an evidence-based approach. 

### 5.1. They “Felt Hot at Home” but Have No Fever Now—Should I Worry?

This is a common conundrum in the emergency department (ED). Does the baby who felt hot at home really have a fever and do they really need further investigation? Parental reports of fever should be taken seriously. In the UK, NICE recommends that parental perceptions of fever be considered valid [8]. This recommendation is supported by a large study of 2470 young febrile infants that found that rates of IBI were exactly the same in those with fever at home (16 had IBI from 678 = 2.4%) compared with those who were febrile at initial assessment (43 had IBI from 1792 = 2.4%) [28].

Most parents admit using touch initially as a method of detecting fever [29]. Even when parents own a thermometer, half of them will continue to use touch as a way of detecting fever [29]. A study examining the validity of this in infants <3 months of age found that both the specificity and sensitivity of parental perceived fever correlated with true fever when a thermometer was used [30]. In addition, studies have shown that when parents think their child is afebrile, they are also likely to be correct [31].

### 5.2. The Baby’s Urine Dip Is Positive, How Reliable Is This?

Febrile infants have a much higher risk of UTI (9%–17%) compared to IBI (1–3%) [1,2,3,32]. The investigation of infants with suspected UTI has been dependent on obtaining urine for microscopy and culture, which is supported by NICE NG224 (urinary tract infection in under 16s: diagnosis and management) [33]. The results of these investigations can be delayed or take up to 24 h and are not available to the treating clinician at the front door, leading to a treat-all approach for febrile infants with suspected UTI. Rapid point-of-care urinary dipsticks have become readily available in the last decade and are used for older infants and children, as supported by various guidelines [33,34,35].

In older children, the sensitivity and specificity of positive urinalysis (leukocyte esterase (LE) or nitrite positive) are 93% and 72%, respectively [34]. When positive microscopy is used in combination with positive LE or nitrites, this improves the sensitivity to 99.8% with a small drop in specificity to 70% [34]. Recent studies conducted in North America and Spain have shown sensitivities ranging from 84% to 93% and specificities ranging from 92% to 95% in febrile infants with positive LE or nitrite [32,36]. Both studies were secondary analyses of large prospective datasets, using a threshold of ≥50,000 cfu/mL of a single pathogen to define a positive urine culture [32,36]. Both studies collected urine only via the sterile method (catheter sampling or suprapubic aspiration) [32,36]. When positive microscopy (presence of >5 white blood cells (WBCs) per high-power field (HPF)) was included for positive urinalysis (LE or nitrite or microscopy), improved sensitivity (94%) was noted with a minimal decrease in specificity (91%) [37]. 

A study conducted in the UK and Ireland across six paediatric emergency departments showed a sensitivity of 84% and specificity of 71% for positive LE or nitrites [38]. These findings were lower than those in the aforementioned studies conducted in North America and Spain. The difference noted was in the urine collection methods, with 8% of samples being collected via a catheter compared to 92% using the clean-catch method [38]. A threshold of ≥100,000 cfu/mL for a single pathogen was used [38]. For all three studies, the LE was the most sensitive test when it came to ruling out UTI while nitrite was the most specific and hence useful for ruling in UTI [35,36,37,38]. These studies show that urinalysis (LE and nitrite) is a good screening test for UTI and can be used in combination with microscopy. These findings are comparable to the diagnostic accuracy of point-of-care urinalysis for older children and support its use in febrile infants younger than three months [34]. However, it is important to note that the accuracy of urinalysis in this cohort is likely to decrease when nonsterile methods of urine collection are employed. 

### 5.3. Do Infants with Urinary Tract Infection Routinely Require a Lumbar Puncture to Exclude Meningitis?

Lumbar puncture is an unpleasant procedure, and data from the UK and Ireland suggest that the average infant undergoes a median of three lumbar puncture attempts before a sample is obtained [13]. Pragmatically, any child who clinically appears to have meningitis or is less than 28 days of age should undergo lumbar puncture to exclude bacterial meningitis. This is supported by the fact that the majority of meningitis cases are detected in neonates [2]. Older infants who otherwise appear well and have a positive urine dipstick or microscopy, suggestive of UTI, do not routinely require a lumbar puncture to exclude bacterial meningitis [39,40]. This approach is supported by a recent PECARN study of 7180 febrile infants aged under 60 days [39]. In that study, the authors reported that no infants aged over 28 days with suspected UTI (n-697) had bacterial meningitis [39]. This result is mirrored by other similar studies, including a recent meta-analysis that included 25,374 infants, and showed that in well-appearing infants aged 29 to 60 days, the occurrence of bacterial meningitis was no higher in those with a positive urinalysis (0.25–0.44%) than in infants with a negative urinalysis (0.28–0.50%) [40,41]. 

### 5.4. Do I Need to Worry about Fever in an Infant following Vaccination?

The UK vaccination schedule has included vaccination against meningococcal B in infants aged between two and four months since 2015 [42]. Meningococcal B vaccination has been shown to cause a fever of >38.5 °C in 50–60% of infants; as a result, parents are advised to prophylactically administer paracetamol following vaccination [42]. The adverse effects of the meningococcal B vaccine can often mimic those of IBIs, including irritability (71%) and reduced feeding (63%) [42]. With so many infants expected to have fever following meningococcal B vaccination, do we need to worry about fever following vaccination?

A study examining infants (n = 35) presenting to the ED within 72 h of meningococcal B vaccination found that 54% underwent blood culture testing, 17% underwent a lumbar puncture, 80% underwent urinalysis testing, and 51% were admitted [43]. The median time to presentation to the ED was 10 h post vaccination, and the median time of fever post vaccination was 7 h [43]. All 35 infants were diagnosed with vaccine-induced fever without IBI [43]. For those presenting more than 24 h post vaccination, the risk of UTI seems to increase. This was demonstrated in a retrospective study of 213 febrile infants aged 6–12 weeks presenting with post-immunisation fever [44]. The prevalence of UTI was 0.6% in patients who presented within 24 h [44]. This increased to 8.9% when presenting after 24 h [44]. Of note, this study predates the addition of the meningococcal B vaccination and includes the time when Pediarix, a pentavalent vaccine, was introduced. It is useful to note that other vaccine-induced fevers also yield similar results. A more recent study conducted after the introduction of meningococcal B included 185 patients who presented with fever post vaccination [45]. Of these, 83.3% (n = 155) received the hexavalent vaccine, which included meningococcal B. UTI was the only SBI found in this recently vaccinated cohort, and there were no cases of IBI [45]. 

With rates of IBI approaching zero in infants presenting within 12–24 h of meningococcal B vaccination, it is reasonable to limit investigations, withhold antibiotics, and observe. Urinalysis will help to exclude occult UTI and should be considered. Blood testing is of limited value, with CRP levels frequently elevated following meningococcal B vaccination (median CRP value 25.9 mg/L) [46].

### 5.5. Should a Positive Viral Respiratory Swab Alter the Management of Febrile Infants?

Viral testing has improved over the last two decades, with rapid pathogen identification occurring within 24 h of testing [47]. Rapid viral testing has been increasingly embedded within paediatric care models to aid in the diagnosis and management of febrile children. However, what are the implications of a positive viral test on the likelihood of IBI and UTI, and should this change management? 

Large studies from North America have shown that the likelihood of IBI in febrile infants with a positive viral test is less than 1.2% compared with 3.7% in the virus-negative group [48,49]. The most commonly detected viruses are Rhinovirus, Enterovirus, and Respiratory Syncytial Virus (RSV) among young febrile infants, and, more recently, SARS-CoV-2 [48,49,50]. A systematic review of infants under 60–90 days admitted with RSV bronchiolitis identified 11 relevant studies and showed that there were no cases of IBI in children with RSV bronchiolitis, although 3.3% were found to have a concurrent UTI [51]. Similarly, a large North American study (n = 14,402) of infants with SARS-CoV-2 reported that UTIs occurred in less than 1% and IBI occurred in approximately 0.3% [52]. 

Interpreting and applying these data to clinical practice is challenging, as a non-negligible risk of IBI and UTI remains, even in the presence of a viral pathogen. For infants aged 28 days and under or who are unwell, there is likely to be no role for viral testing in excluding possible IBI. For those who appear well, infants over 28 days of age with rapid viral testing could potentially be included in the sequential assessment to reduce the need for blood testing and/or lumbar puncture. This could also facilitate early discharge if the infant is admitted to the hospital for a period of observation.

Now that we have worked through our five conundrums, we wanted to summarise our findings in Table 1. 

## 6. Future Research

Over of the past two decades, research teams in Europe and the USA have worked to refine the assessment and management of febrile infants [2,4,10]. This has led to the development of tailored care pathways that can identify small groups of infants at the lowest risk of IBI. Refining these pathways further will be challenging. Two areas where further development could yield significant results are in the assessment of infants with a very short history of fever and infants with abnormal urinalysis. 

Febrile infants often present early in their illness and there is evidence that current guidelines and biomarkers perform poorly in those infants with the shortest duration of fever [53,54]. Research to develop a tailored approach specifically for this group could help reduce cases of missed IBI. Similarly, biomarker discovery focusing on RNA-based biomarkers could further improve tailored care [55]. Unlike protein biomarkers, which take several hours to rise in response to infection, RNA-based markers are often elevated within minutes. A stable RNA marker that rises early in response to infection can revolutionise care for febrile infants. Several UK and European trials are ongoing to investigate the role of different RNA signatures of bacterial infections in children of different ages [56,57,58]. 

Infants with abnormal urinalysis also represent a significant challenge. The presence of pyuria or bacteriuria may represent urinary tract infection, infection elsewhere, or urine contamination. Faced with this dilemma, clinicians must decide whether to treat suspected UTI or observe pending urine culture results. Another potential approach, as advocated in the American Academy of Paediatrics guidance, is to treat these children empirically with oral antibiotics pending urine cultures [2]. Concerns remain, however, that this approach could lead to partially treated UTIs and progression to IBI. The Empirical Oral AntibioticS for possible UTI in well-appearing Young febrile infants (EASY) trial is an NIHR-funded trial due to open in 2024 that will randomise well-appearing febrile infants with abnormal urinalysis and suspected UTI to treatment with either intravenous or oral antibiotics. If oral antibiotics are found to be noninferior to intravenous antibiotics in this cohort, then a much greater number of infants can be managed safely in the community. 

## 7. Conclusions

Febrile young infants are at higher risk of IBI and UTI than older children and can be challenging to assess. This has historically led to a cautious approach, with many infants receiving an extensive diagnostic workup and parenteral antibiotics “just in case”. In high-resource settings with good antenatal care, adopting a tailored approach based on sequential assessment is safe. This approach has many benefits, including lower healthcare costs, better antimicrobial stewardship, and fewer painful procedures. With any risk-based approach there will be some risk. These risks can be mitigated and quantified, but, ultimately, the application of tailored care relies on the views of the clinical team, communication with caregivers, and effective shared care. 

## Figures and Tables

**Figure 1 antibiotics-13-00088-f001:**
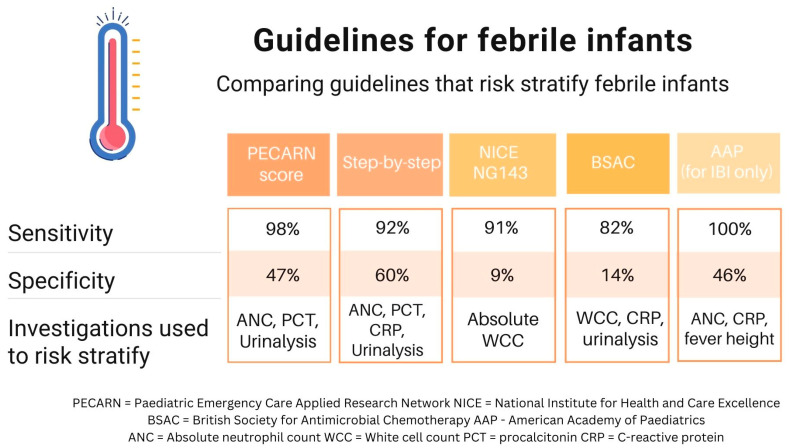
Comparing guidelines that risk stratify febrile infants.

**Table 1 antibiotics-13-00088-t001:** Five common conundrums and responses from evidence review.

Conundrum	Response
They “felt hot at home” but have no fever now—should I worry?	Infants with measured fever at home are still at risk of IBI even if afebrile in the emergency department.
The baby’s urine dip is positive, how reliable is this?	Urinalysis is reliable screening test for UTI but should be interpreted in line with the method of urine collection.
Do infants with urinary tract infection routinely require a lumbar puncture to exclude meningitis?	Infants with urinary tract infection and low risk for IBI do not require routine lumbar puncture.
Do I need to worry about fever in an infant following vaccination?	Infants post vaccination may only require observation and urinalysis for their evaluation in the emergency department if they present with fever.
Should a positive viral respiratory swab alter the management of febrile infants?	Infants with positive viral swab have lower risk of IBI and UTI. May still require investigation based on age and clinical appearance.

## Data Availability

Not applicable.

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
