# Peer review of "Conundrums in the Management of Febrile Infants under Three Months of Age and Future Research"

_antibiotics, 2024, doi:10.3390/antibiotics13010088_

Round 1
Reviewer 1 Report
Comments and Suggestions for Authors This is a narrative review of common clinical questions in the management of pediatric patients presenting with fever, which aims to address 5 key clinical questions, with the lens of stewardship opportunities. It is relevant particularly through the stewardship lens. This is a narrative review, it does not add new data but it does summarize multiple data sources (studies, guidelines) succinctly. Not relevant to narrative review. And the conclusions are consistent with the evidence and arguments, the references are appropriate.Please add search engines and terms to the introduction
could the authors assign an overall grade level to the evidence for each question
The table is nice, consider adding another table that summarizes the key 5 questions, a short summary of the text, and the “answer”
also figures that depict the diagnostic pathways and concerted treatment (or not) for low, standard, and high risk be a very helpful addition
Author Response
Thank you very much for taking the time to review this manuscript. We have made the following changes based on your suggestions.
Comments 1: Please add search engines and terms to the introduction
Response 1: Thank you for pointing this out. We agree with this comment. Therefore, we have added a paragraph in the introduction to include our search engines and terms (lines 55 to 64).
Comments 2: Could the authors assign an overall grade level to the evidence for each question
Response 2: Thank you for this comment. As this is a narrative review and not systematic review or guideline we did not undertake quality assessment for each paper or provide grade recommendation for each conundrum. This is beyond the scope of this narrative review.
Comment 3: The table is nice, consider adding another table that summarizes the key 5 questions, a short summary of the text, and the “answer”.
Response 3: Thank you for this comment, as a result we have added a table to summarise the 5 clinical conundrums and the response.
Comment 4: ..also figures that depict the diagnostic pathways and concerted treatment (or not) for low, standard, and high risk be a very helpful addition
Response 4: We agreed that this would be valuable, however, as discussed within the article, the diagnostic pathway and treatment is dependent on country and resources. It is therefore too complex to simplify into a diagram.
Reviewer 2 Report
Comments and Suggestions for Authors
In this review, the authors have identified five conundrums in managing fabrile infants based on comparative analysis between various guidelines. However, authors are requested to address the following issues to improve the overall quality of the manuscript.
1. The article needs to be structured, i.e., introduction, method, etc.
2. Subheadings need to be numbered.
3. Figures and tables must be numbered and appropriately mentioned in the manuscript.
4. The figure should contain proper legends, and each abbreviation must be elaborately mentioned in the legend and the manuscript.
5. The solutions for each conundrum need to be discussed based on established guidelines and comparatively addressed using appropriate tables and, if possible, with statistical justifications.
Comments on the Quality of English Language
The manuscript must be thoroughly checked for typos and formatting errors.
Author Response
Thank you very much for taking the time to review this manuscript. Please find the detailed responses below.
Comment 1: The article needs to be structured, i.e., introduction, method, etc.
Response 1: Many thanks for this comment. We have added a section within the introduction that details our methods including how we conducted our literature review. This article is a narrative review that discusses multiple topics.
Comment 2: Subheadings need to be numbered.
Response 2: Many thanks for highlighting this, we have numbered all subheadings.
Comment 3: Figures and tables must be numbered and appropriately mentioned in the manuscript.
Response 3: Many thanks for noticing this, we have introduced each table and appropriately numbered each table.
Comment 4: The figure should contain proper legends, and each abbreviation must be elaborately mentioned in the legend and the manuscript.
Response 4: Many thanks for noticing this, we have altered the figure to contain a legend with each abbreviation mentioned.
Comment 5: The solutions for each conundrum need to be discussed based on established guidelines and comparatively addressed using appropriate tables and, if possible, with statistical justifications.
Response 5: Many thanks for this. Each conundrum has been discussed in line with the American Academy of Pediatric guidance and the National Institute for Care Excellence guidelines for fever under 5. Given that this is a narrative review, we do not feel it lends itself to statistical analysis.
Reviewer 3 Report
Comments and Suggestions for Authors
Dear authors,
I have read your paper with interest and I have the following comments:
MAJOR COMMENTS
-despite the authors highlight that some guidelines have a low threshold for antibiotic treatment of specific paediatric age groups presenting with fever, no reflection on the potential influence on antimicrobial resistance is being made. This should be added in my view
Minor comments
-some acronyms should be explained in the text (e.g. SBI) and this should be done at first appearance (e.g. PECARN).
Author Response
Thank you very much for taking the time to review this manuscript. Please find the detailed responses to your comments below:
Comment 1: Despite the authors highlight that some guidelines have a low threshold for antibiotic treatment of specific paediatric age groups presenting with fever, no reflection on the potential influence on antimicrobial resistance is being made. This should be added in my view
Response 1: Many thanks for this valuable point. We agree with your comment and as such have added lines 148-152 that talks about the need for antimicrobial stewardship to limit antimicrobial resistance.
Comment 2: Some acronyms should be explained in the text (e.g. SBI) and this should be done at first appearance (e.g. PECARN).
Response: Many thanks for highlighting this, we have adjusted the manuscript so that acronyms are explained when they first appear.
Round 2
Reviewer 2 Report
Comments and Suggestions for Authors
In the manuscript, the authors tried to address the comments made in the first round of the review. However, the authors are requested to address the following minor issues carefully:
1. There should be a separate method section.
2. Authors are requested not to add headings as questions. For example, “Why is there a difference between practices worldwide and what clinical practice guidelines are available?” Rather, direct sentences are suggested, like Clinical practice guidelines and differences.
3. Table 1 is not cited in the body.
Comments on the Quality of English LanguageMinor editing of the English language is required
Author Response
Thank you very much for taking the time to review this manuscript. Please find the detailed responses below.
Comment 1: There should be a separate method section.
Response 1: Thank you for highlighting this. We have now added a separate methods section.
Comment 2: Authors are requested not to add headings as questions. For example, “Why is there a difference between practices worldwide and what clinical practice guidelines are available?” Rather, direct sentences are suggested, like Clinical practice guidelines and differences.
Response 2: Thank you, we agree with you comment and have changed headings 3 and 4 accordingly. However, with regards to our 5 clinical conundrums, we have kept these as questions as that was the original narrative of the article.
Comment 3: Table 1 is not cited in the body.
Response 3: Many thanks, we have added sentence 155 and 156 as a result of your comment.
Reviewer 3 Report
Comments and Suggestions for Authors
No further comments
Author Response
Thank you